# miR-194 Accelerates Apoptosis of Aβ_1–42_-Transduced Hippocampal Neurons by Inhibiting Nrn1 and Decreasing PI3K/Akt Signaling Pathway Activity

**DOI:** 10.3390/genes10040313

**Published:** 2019-04-21

**Authors:** Tingting Wang, Yaling Cheng, Haibin Han, Jie Liu, Bo Tian, Xiaocui Liu

**Affiliations:** Psychiatric Department V, Qingdao Mental Health Center, No. 299, Nanjing Road, Shibei District, Qingdao 266000, China; wtt_920@163.com (T.W.); ling821118@sohu.com (Y.C.); hanhaibinqf@126.com (H.H.); jiejie19890128@126.com (J.L.); boyangqd@163.com (B.T.)

**Keywords:** Alzheimer’s disease, hippocampal neurons, miR-194, Nrn1, proliferation, apoptosis, PI3K/AkT signaling pathway

## Abstract

This article explores the mechanism of miR-194 on the proliferation and apoptosis of Aβ_1–42_-transduced hippocampal neurons. Aβ_1–42_-transduced hippocampal neuron model was established by inducing hippocampal neurons with Aβ_1–42_. MTT assay and flow cytometry were used to detect the viability and apoptosis of hippocampal neurons, respectively. qRT-PCR was used to detect changes in miR-194 and Nrn1 expression after Aβ_1–42_ induction. Aβ_1–42_-transduced hippocampal neurons were transfected with miR-194 mimics and/or Nrn1 overexpression vectors. Their viability and neurite length were detected by MTT assay and immunofluorescence, respectively. Western blot was used to detect protein expression. Aβ_1–42_ inhibited Aβ_1–42_-transduced hippocampal neuron activity and promoted their apoptosis in a dose-dependent manner. miR-194 was upregulated and Nrn1 was downregulated in Aβ_1–42_-transduced hippocampal neurons (*p* < 0.05). Compared with the model group, Aβ_1–42_-transduced hippocampal neurons of the miR-194 mimic group had much lower activity, average longest neurite length, Nrn1, p-AkT, and Bcl-2 protein expression and had much higher Bax, Caspase-3, and Cleaved Caspase-3 protein expression. Compared with the model group, Aβ_1–42_-transduced hippocampal neurons of the LV-Nrn1 group had much higher activity, average longest neurite length, Nrn1, p-AkT, and Bcl-2 protein expression and had much lower Bax, Caspase-3, and Cleaved Caspase-3 protein expression. Nrn1 is a target gene of miR-194. miR-194 inhibited apoptosis of Aβ_1–42_-transduced hippocampal neurons by inhibiting Nrn1 and decreasing PI3K/AkT signaling pathway activity.

## 1. Introduction

Alzheimer’s disease (AD) is an age-related neurodegenerative disease with clinical features such as progressive cognitive impairment and psychomotor disorders [1,2]. The pathogenesis of Alzheimer’s disease involves a variety of factors, including genetic factors, vascular disease, and malnutrition [3,4,5]. According to statistics, about 27 million people are living with AD [6]. AD is mainly characterized by the formation of insoluble senile plaques [7]. β-amyloid (Aβ), a cleavage product of the amyloid precursor protein (APP), is the main component of insoluble senile plaques, whose deposition is neurotoxic and could induce apoptosis of neurons [8,9,10]. AD has a serious negative impact on patients’ quality of life. The prevention and treatment of AD are still a major clinical problem all over the world. The discovery of exact molecular pathogenesis has important clinical implications for AD patients.

miRNA is a class of small molecule noncoding RNA, which has been reported to be involved in the development of various diseases [11]. It has also been found to be widespread in the brain and related to cell cycle and apoptosis [6]. In AD, several miRNAs have been discovered to be abnormally expressed. Jia et al. [12] recruited 84 AD patients and 62 healthy volunteers for serum miRNA level detection. Their results indicated that, compared with healthy volunteers, AD patients had much lower serum levels of miR-29, miR-125b, and miR-223, while their serum miR-519 level was dramatically higher than that of healthy volunteers. Moncini and colleagues [13] pointed out that the miR-15/107 family displays declined expression in the hippocampus and temporal cortex and that the CDK5R1 mRNA is upregulated in AD hippocampus tissues. Further research has indicated that decreased expression of miR-15/107 family miRNAs regulates the pathogenesis of AD by promoting CDK5 expression and enhancing its activity. An et al. [14] reported impaired miR-124 expression in AD, and they also noticed that miR-124 might be involved in AD pathogenesis through negative regulation of BACE1. miR-194 has also been reported to be involved in the development and progression of many diseases, such as tumors and inflammatory reactions. In 2013, Pierre et al. [15] researched alteration of the microRNA network during the progression of Alzheimer’s disease and found that miR-194 is differently expressed in this disease. More importantly, miR-194-5p has been confirmed to be significantly downregulated in AD patients [16]. However, the specific mechanisms of miR-194, such as the related targets and pathways of AD, have not been fully studied.

In this study, an Aβ_1–42_-transduced hippocampal neuron cell model was established by Aβ_1–42_ induction. The effects of miR-194 on Aβ_1–42_-transduced hippocampal neuron activity and apoptosis were explored. Relevant mechanisms were also investigated further in order to provide potential therapeutic target and theoretical basis for AD treatment.

## 2. Material and Methods

### 2.1. Primary Culture of Rat Hippocampal Neurons

Wistar rats (purchased from Shandong Lukang Pharmaceutical Company Experimental Animal Center) aged 24 h were sterilized with 75% ethanol, and the hippocampus tissues were separated under aseptic conditions after they were anesthetized by intraperitoneally injecting with 10% chloral hydrate (dose: 5 mL/kg). The study was approved by the ethics committee of Qingdao Mental Health Center. All efforts were made to reduce animal suffering. The hippocampus tissues were placed in precooled phosphate-buffered saline (PBS) and cut into 1 mm^3^ pieces after removal of blood vessels and fascia. DMEM/F12, containing 15% fetal bovine serum (FBS), was used to suspend hippocampal neurons isolated from the tissues. After centrifugation for 5 min at 1000 r/min, these hippocampal neurons were redispersed in DMEM/F12 (15% FBS) at a density of 2 × 10^6^/mL. Hippocampal neurons were incubated in an incubator at 37 °C, 5% CO_2_, for 24 h. The residual liquid was completely replaced with maintenance medium for 72 h incubation. Then, half of the remaining maintenance medium was discarded and replaced with fresh maintenance medium containing cytarabine (concentration of 2.5 mg/L). Cytarabine can inhibit the growth of fibroblasts and glial cells. Thereafter, half of the maintenance medium was replaced with fresh maintenance medium containing cytarabine (2.5 mg/L) every 3 days. Hippocampal neurons were continuously cultured for 7 days, and their morphology was observed under an inverted microscope at day 1, 2, and 7.

### 2.2. Immunofluorescence Labeling of Hippocampal Neurons

Hippocampal neurons were inoculated in 6-well plates. A coverslip was placed at the bottom of each well before inoculation. After 7 days of incubation, the coverslip was rinsed twice with PBS. Paraformaldehyde (4%) was used to fix hippocampal neurons for 20 min. Nonspecific binding sites were blocked by 10% goat serum albumin for 30 min at 37 °C. Hippocampal neurons were incubated with mouse anti-rat β-III tubulin monoclonal antibody (ab7291, 1:200, abcam, Cambridge, UK) for 12 h at 4 °C. The hippocampal neurons were washed three times with PBS for 10 min each time. Fluorescein isothiocyanate (FITC)-labeled goat anti-mouse IgG secondary antibody (ab6785, 1:50, abcam, Cambridge, UK) was also added for 2 h incubation at room temperature in darkness. The nucleus was counterstained for 1 h using Hoechst 33258 (50 μg/mL, Sigma, Kawasaki City, Japan) at room temperature in darkness. The hippocampal neurons were again washed three times with PBS for 10 min each time. After excess liquid on the coverslip was dried, hippocampal neurons were observed under an inverted fluorescence microscope. In the same field of view, fluorescence pictures of neurons and neuronal nuclei were obtained separately using different excitation lights. Pictures were merged using NIS imaging software. The longest neurite length of hippocampal neurons in each field of view was measured using Image Pro Plus 6.0 software (Media Cybernetics, Maryland, MD, USA).

### 2.3. MTT Assay for Hippocampal Neuron Survival after Induction of Aβ_1–42_

Hippocampal neurons were seeded in 96-well plates at a density of 1 × 10^4^/well with 100 μL of maintenance medium (2.5 mg/L of cytarabine) in each well. After being cultured for 24 h, hippocampal neurons were induced by different concentrations of Aβ_1–42_ (0, 10, 20, 30, 40, and 50 μmol/L). Among them, 0 μmol/L Aβ_1–42_ was considered as the negative control. Aβ_1–42_ peptide and its scrambled sequence (control), synthesized by Shanghai Chu Peptide Biotechnology Co., Ltd., were as follows: Aβ_1–42_, DAEFRHDSGYEVHHQKLVFFAEDVGSNKGAIIGLMVGGVVIA; scrambled Aβ_1–42_, AIAEGDSHVLKEGAYMEIFDVQGHVFGGKIFRVVDLGSHNVA. Aβ_1–42_ oligomers were prepared as previously described [17]. Briefly, Aβ_1–42_ was dissolved in hexafluoroisopropanol to a concentration of 1 mM, incubated for 60 min at room temperature, and the hexafluoroisopropanol was then evaporated. The dried films were reconstituted in dimethyl sulfoxide (DMSO) to a concentration of 5 mM, diluted to 200 μM in sterile PBS containing 0.2% sodium dodecyl sulfate (SDS), and incubated at 4 °C for 24 h. Finally, the above preparation was diluted with PBS to a final concentration of 10 μM. All hippocampal neurons were cultured for 24, 48, and 72 h. MTT solution (5 mg/mL) was added with 20 μL per well for 4 h incubation at 37 °C, followed by replaced with 150 μL DMSO. All 96-well plates were shaken for 10 min to promote complete dissolution of purple crystals. The absorbance values at 570 nm (OD_570_) of each well were detected on a microplate reader. The viability of hippocampal neurons in each well was calculated.

### 2.4. Flow Cytometry for Apoptosis

Hippocampal neurons were collected after 7 days induction with different concentrations of Aβ_1–42_ (0, 10, 20, 30, 40, and 50 μmol/L). After being stained with Annexin V-FITC and propidium iodide (PI), apoptosis detection was performed using flow cytometry.

### 2.5. qRT-PCR

Total RNA from normal hippocampal neurons (blank group) and hippocampal neurons induced by 30 μmol/L Aβ_1–42_ (Aβ_1–42_ group) were extracted by TRIzol reagent (Thermo Fisher Scientific, Waltham, MA, USA). A total of 0.5 mg RNA was subjected to the synthesis of cDNA by reverse transcription reaction. Fast Start Universal SYBR Green Mastermix (Roche, Basel, Switzerland) was selected to conduct qRT-PCR according to the following conditions: predenaturation at 95 °C for 10 min, denaturation at 95 °C for 10 s, annealing at 60 °C for 20 s, and extension at 72 °C for 34 s. This process included 40 cycles. Primers used in this research were as follows: miR-194, forward, TCCGAAGGTGTACCTCAAC, reverse, GTGCAGGGTCCGAGGT; U6, forward, CTCGCTTCGGCAGCACATATACT, reverse, ACGCTTCACGAATTTGCGTGTC; Nrn1, forward, AGCATGGCCAACTACCCGCA, reverse, CCGCTGCCGCAGAGTTCGAATA; GAPDH, forward, GGTGCTGAGTATGTCGTGGAGT, reverse, CAGTCTTCTGAGTGGCAGTGATG. miR-194 expression was normalized to U6, and Nrn1 mRNA expression was normalized to GAPDH. Data was processed by the 2^−ΔΔCt^ method.

### 2.6. Luciferase Reporter Gene Assay

Based on the prediction results of Target Scan, the 3′UTR fragment of Neuritin (Nrn1) was cloned and ligated into the pmir GLO plasmid (Promega, Madison, WI, USA) to construct wild-type reporter vector (pmir GLO-Nrn1-Wt). Meanwhile, the mutant sequence of the Nrn1 3′UTR fragment was also obtained using MutanBEST Kit (TaKaRa, Japan), followed by being ligated into the pmir GLO plasmid to construct mutant-type reporter vector (pmir GLO-Nrn1-Mut). Normal hippocampal neurons were subjected to transfection by miR-194 mimics or its negative control (NC). The sequences of miR-194 mimics and its negative control (RiboBio Co., Guangzhou, China) were as follows: miR-194 mimics, 5′-UGUAACAGCAACUCCAUGUGGA-3′; miR-194 mimic negative control, 5′-UUCUCCGAACGUGUCCGGAGAATT-3′. Then, transfection by pmir GLO-Nrn1-WT or pmir GLO-Nrn1-Mut was further performed on these hippocampal neurons. The hippocampal neurons were grouped as follows depending on the transfection sequences: NC + Wt group, mimics + Wt group, NC + Mut group, and mimics + Mut group. All transfection operations were carried out in strict accordance with the Lipofectamine^®^ 2000 instructions (Thermo Fisher Scientific, Waltham, MA, USA). The successfully transfected hippocampal neurons were placed in the incubator for 48 h at 37 °C, 5% CO_2_. Their luciferase activity was tested using Dual Luciferase Activity Assay Kit (Promega, Madison, WI, USA).

### 2.7. Hippocampal Neuron Transfection

Aβ_1–42_-transduced hippocampal neurons induced by 30 μmol/L of Aβ_1–42_ were collected and transfected by miR-194 mimics (miR-194 mimic group) or Nrn1 overexpression vectors (LV-Nrn1 group). In addition, they were also cotransfected with both miR-194 mimics and Nrn1 overexpression vectors (lentiviral backbone) and were served as the miR-194 mimics + LV-Nrn1 group. Lentiviral vectors were purchased from Tronolab, and the packaging cell 293T cell line was purchased from the Shanghai Institute of Cellular Sciences, Chinese Academy of Sciences. Furthermore, Aβ_1–42_-transduced hippocampal neurons only induced by 30 μmol/L of Aβ_1–42_ were set as the model group. miR-194 mimics and Nrn1 overexpression vectors were purchased from Guangzhou Ruibo Biotechnology Co., Ltd., China. Transfection was performed in strict accordance with the instructions of Lipofectamine^®^ 2000. Briefly, hippocampal neuron cells were plated in 6-well plates at a density of 5 × 10^5^/well and allowed to adhere for 24 h. An amount of 5 μL Lipofectamine 2000 with miR-194 mimics and/or Nrn1 overexpression vectors were used per well. After transfection, all Aβ_1–42_-transduced hippocampal neurons were cultured in the 37 °C, 5% CO_2_ incubator for 0–72 h. Cell viability was also detected using MTT assay as above.

### 2.8. Detection of Mitochondrial Transmembrane Potential

Residual liquid in each well was removed after the hippocampal neurons were incubated for 7 days with different concentrations of Aβ_1–42_ (0, 10, 20, 30, 40, and 50 μmol/L). A total of 1 mL Rhodamine 123 fluorescent probe (10 μM) was added into each well for 20 min incubation at 37 °C in darkness. It was necessary to wash with PBS three times to remove residual probes. Hippocampal neurons were observed under an inverted microscope.

### 2.9. Western Blot Analysis

Hippocampal neurons of each group were lysed with radioimmunoprecipitation assay (RIPA) buffer (Thermo Fisher Scientific, Waltham, MA, USA) to get total proteins. Bicinchoninic acid (BCA) kit (Promega, Madison, WI, USA) was used to quantify these proteins. After being subjected to sodium dodecyl sulfate polyacrylamide gel electrophoresis (SDS-PAGE), proteins were transferred onto polyvinylidene difluoride (PVDF) membrane. Primary antibodies used to blot proteins were rabbit anti-human Nrn1 antibody (bs-2464R, 1:1000, Bioss, Boston, MA, USA), rabbit anti-human Akt (#4691, 1:1000, Cell Signaling Technology, Boston, MA, USA), rabbit anti-human p-Akt (#4060, 1:1000, Cell Signaling Technology, Boston, MA, USA), rabbit anti-human Bax (ab199677, 1:1000, abcam, Cambridge, UK), rabbit anti-human Bcl-2 (ab196495, 1:1000, abcam, Cambridge, UK), rabbit anti-human Caspase-3 (#9662, 1:1000, Cell Signaling Technology, Boston, MA, USA), and rabbit anti-human Cleaved Caspase-3 (#9661, 1:1000, Cell Signaling Technology, Boston, MA, USA). The membrane was incubated at 4 °C for 12 h and then washed with tris-buffered saline (TBST) 3 times for 10 min each time. Horseradish peroxidase-labeled goat anti-rabbit IgG (#sc-2004, 1:5000; Santa Cruz Biotechnology, Dallas, TX, USA) was used as secondary antibody to perform 1 h incubation at room temperature. Enhanced chemiluminescence system was used to visualize immunoreactive bands.

### 2.10. Statistical Analysis

All data were processed using SPSS 21.0 statistical software. Measurement data are expressed as mean ± standard deviation (SD). The two-tailed *t*-test was used for comparison between two groups, while one-way analysis of variance was selected for comparison among multiple groups. *p* < 0.05 indicated that the difference was statistically significant. All experiments were repeated at least 3 times in this research.

## 3. Results

### 3.1. Culture and Identification of Hippocampal Neurons

Immediately after inoculation, hippocampal neurons were evenly distributed, round, small, and translucent (Figure 1A). After 1 day of culture, neurons attached to the bottom of the flask, and some of them had already developed projections (Figure 1B). On the next day, larger vertebral-shaped neurons were found. The axons of the neurons began to cross-contact with the adjacent neurons (Figure 1C). Until the 7th day, interconnections between neuron dendrites formed a dense network, and neurons were plump and mature (Figure 1D). After fluorescent labeling with neuron-specific markers, neurons were observed under an inverted fluorescence microscope. Typical hippocampal neurons were cone-shaped, with a long axon and 1–2 dendrites. Branches of dendrites and axons could form cross-densely-connected networks (Figure 1E). Immunofluorescence labeling demonstrated that the purity of hippocampal neurons reached 94.2% ± 3.6%, which could meet the requirements for further experiments (Figure 1F–H).

### 3.2. Aβ_1–42_ Inhibited Hippocampal Neuron Viability and Promoted Its Apoptosis in a Dose-Dependent Manner

After being induced for 24 h, hippocampal neurons treated with 30–50 μmol/L Aβ_1–42_ had much lower viability than the negative controls (*p* < 0.05). At 48 h, hippocampal neuron viability was obviously inhibited when the concentration of Aβ_1–42_ exceeded 20 μmol/L (*p* < 0.05); at 72 h, hippocampal neuron viability was all significantly suppressed after they were treated with 10–50 μmol/L Aβ_1–42_ (*p* < 0.05). It should also be noted that Aβ_1–42_ inhibited the viability of hippocampal neurons in a dose-dependent manner (Figure 2A). In addition, Aβ_1–42_ was able to promote apoptosis of hippocampal neurons in a dose-dependent manner. When hippocampal neurons were treated with 20–50 μmol/L Aβ_1–42_, their apoptosis rates were all dramatically higher than the negative controls (*p* < 0.05) (Figure 2B). An amount of 30 μmol/L Aβ_1–42_ could inhibit hippocampal neuron viability and promote its apoptosis after being induced for 24 h. Therefore, the dose was used in subsequent studies.

### 3.3. Upregulation of miR-194 and Downregulation of Nrn1 mRNA in Aβ_1–42_-Transduced Hippocampal Neurons

After being induced by 30 μmol/L Aβ_1–42_ for 48 h, the relative levels of miR-194 in Aβ_1–42_-transduced hippocampal neurons of the Aβ_1–42_ group was much higher than that of the blank group (*p* < 0.05). The scrambled peptide was used for the blank group. In contrast, Nrn1 mRNA was markedly increased in Aβ_1–42_-transduced hippocampal neurons of the Aβ_1–42_ group when compared with that of the blank group (*p* < 0.05) (Figure 3).

### 3.4. Upregulated miR-194 Inhibited Hippocampal Neuron Viability and Promoted Its Apoptosis

After 24–72 h of transfection, compared with the model group, the viability of Aβ_1–42_-transduced hippocampal neurons in the miR-194 mimic group was significantly decreased (*p* < 0.05, Figure 4A). In addition, the apoptosis rate was dramatically higher than that treated with miR-194 mimic transfection (*p* < 0.05, Figure 4B). These results indicated that the expression of miR-194 directly affected the proliferation and apoptosis of Aβ_1–42_-transduced hippocampal neurons.

### 3.5. Nrn1 Is A Target Gene of miR-194

Target Scan is an online prediction software to predict the binding site of miRNAs and genes. According to the database, miR-194 is directly bound to the 3′-UTR region of Nrn1 (Figure 5A). Luciferase reporter gene assay was further conducted to verify the above speculation. The results showed that hippocampal neurons of the mimics + Wt group had much lower luciferase activity than that of the NC + Wt group (*p* < 0.05). There was no significant difference in luciferase activity between the mimics + Mut group and the NC + Mut group (Figure 5B). It is suggested that miR-194 could downregulate Nrn1 expression by targeting the 3′UTR region of Nrn1.

### 3.6. Nrn1 Repaired the Damage of miR-194 on Aβ_1–42-_Transduced Hippocampal Neuron Viability

After 24–72 h of transfection, compared with the model group, the viability of Aβ_1–42_-transduced hippocampal neurons in the miR-194 mimic group was significantly decreased (*p* < 0.05). Meanwhile, significantly increased viability of Aβ_1–42_-transduced hippocampal neurons in the LV-Nrn1 group was found when compared with the model group (*p* < 0.05). However, there was no obvious difference in the viability of Aβ_1–42_-transduced hippocampal neurons between the miR-194 mimics + LV-Nrn1 group and the model group (Figure 6). These results indicated that Nrn1 could repair the damage of miR-194 on Aβ_1–42_-transduced hippocampal neuron viability.

### 3.7. Nrn1 Reversed the Inhibitory Effect of miR-194 on Neurite Growth of Aβ_1–42_-Transduced Hippocampal Neurons

The average longest neurite length was measured. It could be noticed that, compared with the model group, the average longest neurite length was significantly decreased in the miR-194 mimic group (*p* < 0.05) and remarkably increased in the LV-Nrn1 group (*p* < 0.05). No significant difference was found in the average longest neurite length between the miR-194 mimics + LV-Nrn1 group and the model group (Figure 7). Nrn1 reversed the inhibitory effect of miR-194 on neurite growth of Aβ_1–42_-transduced hippocampal neurons.

### 3.8. miR-194 Regulated Apoptosis-Related Protein Expression in Aβ_1–42_-Transduced Hippocampal Neurons by Inhibiting the Activity of PI3K/Akt Signaling Pathway

The PI3K/AkT signaling pathway is one of the major cellular survival signaling pathways, which is involved in the formation of neuronal synaptic spines and synapses [18]. We examined the effect of miR-194 on the PI3K/AkT signaling pathway, as well as the expression of apoptosis-related factors, in Aβ_1–42_-transduced hippocampal neurons. The results demonstrated that, compared with the model group, Aβ_1–42_-transduced hippocampal neurons of the miR-194 mimic group had much lower p-AkT, Nrn1, and Bcl-2 protein expression (*p* < 0.05) and markedly higher Bax, Caspase-3, and Cleaved Caspase-3 protein expression (*p* < 0.05). In contrast, dramatically increased p-AkT, Nrn1, and Bcl-2 protein expression (*p* < 0.05) and remarkably decreased Bax, Caspase-3, and Cleaved Caspase-3 protein expression (*p* < 0.05) was found in the LV-Nrn1 group when compared with the model group. However, no significant difference was found in the protein expression between the miR-194 mimics + LV-Nrn1 group and the model group (Figure 8A–D). These results indicated that miR-194 regulated apoptosis-related protein expression in Aβ_1–42_-transduced hippocampal neurons by targeting the inhibition of Nrn1 expression and decreasing PI3K/AkT signaling pathway activity.

## 4. Discussion

It has been reported that Aβ_1–42_ is used to establish the AD cell model [17,19]. In this study, primary hippocampal neurons were successfully isolated and cultured. Aβ_1–42_ inhibited hippocampal neuron activity and promoted their apoptosis in a dose-dependent manner. Aβ_1–42_ at a density of 30 μmol/L could successfully induce the AD cell model. miR-194 was upregulated and Nrn1 was downregulated in Aβ_1–42_-transduced hippocampal neurons. Further in-depth research indicated that miR-194 accelerated apoptosis of hippocampal neurons in Alzheimer’s disease by targeting the inhibition of Nrn1 and decreasing PI3K/Akt signaling pathway activity.

miR-194 is frequently reported to play an important regulatory role in a variety of tumor diseases, including glioblastoma multiforme, laryngeal squamous cell carcinoma, gastric cancer, and so on [20,21,22]. It exerts an inhibitory effect on the progression of most tumors. This study first studied the effects of miR-194 on the biological behavior of Aβ_1–42_-transduced hippocampal neurons, and the results suggested that miR-194 suppressed Aβ_1–42_-transduced hippocampal neuron activity and accelerated their apoptosis by targeting the inhibition of Nrn1 expression. Nrn1 was first discovered and reported in the visual cortex of rats by Nedivi et al. [23]. It is a neurotrophic factor, which is closely related to the nervous system plasticity and development [24,25]. Nrn1 can stimulate rapid growth and branching of neuritis, which plays an important role in regeneration and repair after the nervous system is damaged [26,27]. Zhang et al. [28] researched the effects of Nrn1 on early brain injury through rat experiments and found that the application of exogenous Nrn1 could significantly alleviate brain cells apoptosis. Their results further indicated that Nrn1 protects neurons by attenuating brain edema and brain cell apoptosis. Liu et al. [29] established a rat model of traumatic brain injury and discovered that Nrn1 could improve neurological scores of rats and repair their damaged neurons. They also revealed that Nrn1 could protect cortical neurons from apoptosis by inhibiting Caspase-3 expression. In this study, miR-194 dramatically suppressed the activity and neurite growth of Aβ_1–42_-transduced hippocampal neurons. In contrast, Nrn1, which is a target inhibited by miR-194, had a reversal effect on Aβ_1–42_-transduced hippocampal neurons. It could promote Aβ_1–42_-transduced hippocampal neuron activity and neurite growth, which further confirmed the role of Nrn1 in repair and regeneration of damaged neurons.

PI3K/Akt signaling pathway is involved in the pathogenesis of many diseases, and its activation can promote cell activity, especially tumor cells [30,31]. After being transfected by miR-194 mimics, p-Akt/Akt was markedly decreased in Aβ_1–42_-transduced hippocampal neurons, while overexpression of Nrn1 could stimulate the expression of p-Akt/Akt. These results indicate that miR-194 might inhibit the activity of PI3K/Akt signaling pathway by inhibiting the expression of Nrn1, thereby suppressing the activity of Aβ_1–42_-transduced hippocampal neurons. These results are consistent with previous research. Li et al. [32] found in their study that increased p-Akt/Akt expression could much improve the memory capacity of AD rats. A recent literature also suggested that the behavioral symptoms and pathological progression of mouse AD models were alleviated after the activation of PI3K/Akt signaling pathway [33]. Furthermore, this research also revealed that miR-194 could affect apoptosis-related protein expression, such as inhibition of Bcl-2 and stimulation of Bax, Caspase-3, and Cleaved Caspase-3, by inhibiting the expression of Nrn1. As we know, Bcl-2 is an antiapoptotic protein that inhibits cell apoptosis, while Bax, Caspase-3, and Cleaved Caspase-3 play a role in promoting apoptosis. miR-194 accelerated Aβ_1–42_-transduced hippocampal neuron apoptosis by regulating the expression of these apoptosis-related proteins.

The AD cell model used in this study has extensive historical precedent. Previous studies have reported that the control of ERK signaling might be useful in preventing Aβ_1–42_ oligomer-induced neurotoxicity in the hippocampus [34]. Researchers have also found that EphB2 overexpression protects hippocampal neurons against Aβ_1–42_ oligomer-induced neurotoxicity in the cellular model of AD [17]. However, there are some limitations in this study. The model may not be relevant to sporadic AD.

## 5. Conclusions

In conclusion, this article studied the effects of miR-194 on Aβ_1–42_-transduced hippocampal neurons and related mechanisms. The results demonstrated that miR-194 was upregulated and Nrn1 was downregulated in Aβ_1–42_-transduced hippocampal neurons. miR-194 accelerated apoptosis of hippocampal neurons in Alzheimer’s disease by targeting the inhibition of Nrn1 and decreasing PI3K/Akt signaling pathway activity. It may be used as a potential biomarker for the prevention, diagnosis, and treatment of AD.

## Figures and Tables

**Figure 1 genes-10-00313-f001:**
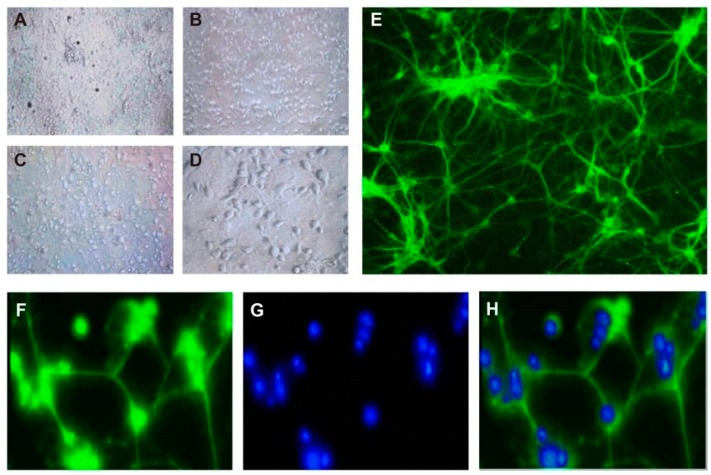
Culture and identification of hippocampal neurons (×200). Morphology of hippocampal neurons (**A**) immediately after inoculation, (**B**) 1 day, (**C**) 2 days, and (**D**) 7 days. (**E**,**F**) Immunofluorescence labeling of hippocampal neurons treated with mouse anti-rat β-III tubulin monoclonal antibody and fluorescein isothiocyanate (FITC)-labeled goat anti-mouse IgG secondary antibody. (**G**) Neuronal nuclei labeled by Hoechst 33258 labeling. (**H**) Merged picture of Figure (**E**,**F**).

**Figure 2 genes-10-00313-f002:**
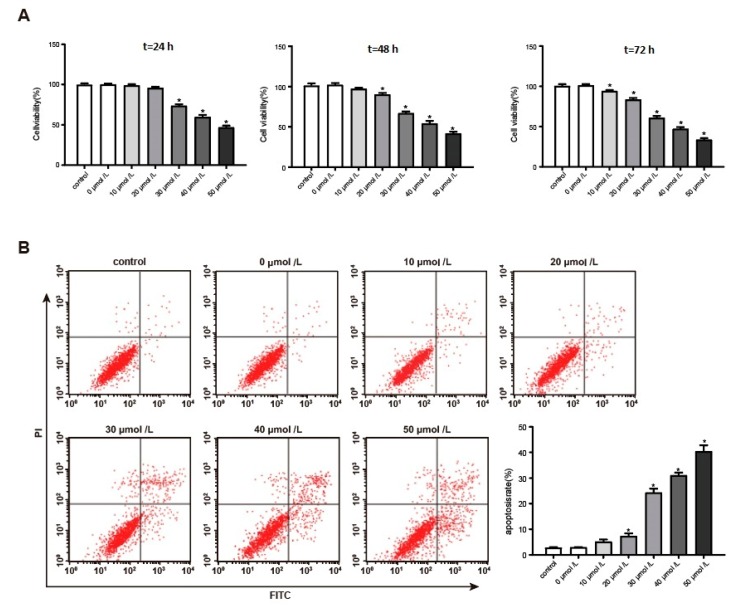
Aβ_1–42_ inhibited hippocampal neuron viability and promoted its apoptosis in a dose-dependent manner. (**A**) MTT assay for hippocampal neuron activity after Aβ_1–42_ induction. (**B**) Detection of hippocampal neuron apoptosis rate by flow cytometry. * *p* < 0.05 when compared with the negative controls.

**Figure 3 genes-10-00313-f003:**
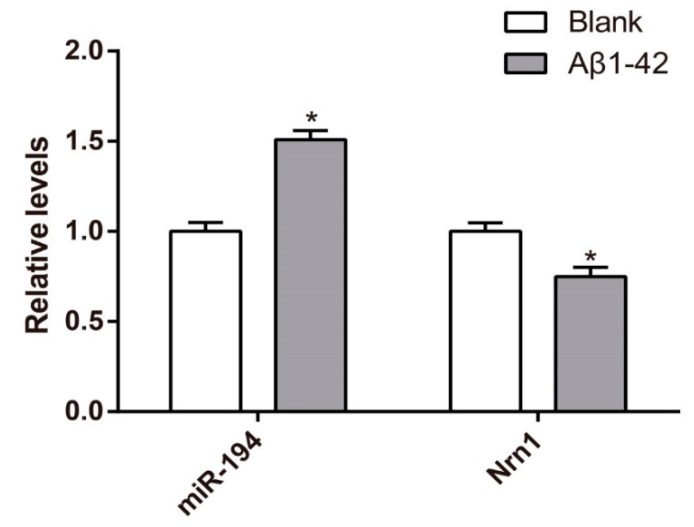
miR-194 was upregulated and Nrn1 mRNA was downregulated in hippocampal neurons after being induced by Aβ_1–42_. * *p* < 0.05 when compared with the blank group.

**Figure 4 genes-10-00313-f004:**
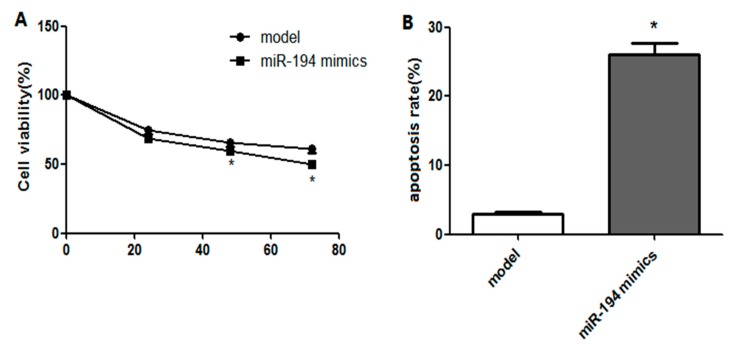
Upregulated miR-194 decreased viability of Aβ_1–42_-transduced hippocampal neurons (**A**) and increased the apoptosis of Aβ_1–42_-transduced hippocampal neurons (**B**). * *p* < 0.05 when compared with the model group.

**Figure 5 genes-10-00313-f005:**
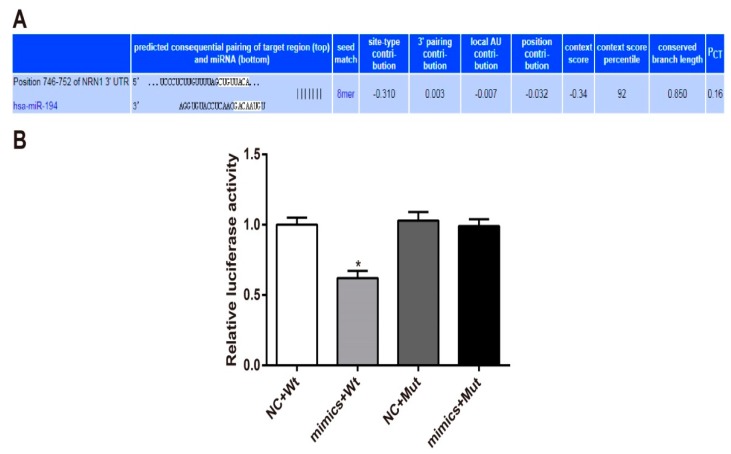
Nrn1 is the target gene of miR-194. (**A**) Target Scan predicted the binding sites of Nrn1 and miR-194. (**B**) Luciferase reporter gene assay was used to verify the targeting relationship between miR-194 and Nrn1. * *p* < 0.05 when compared with the negative control (NC) + wild-type (Wt) group.

**Figure 6 genes-10-00313-f006:**
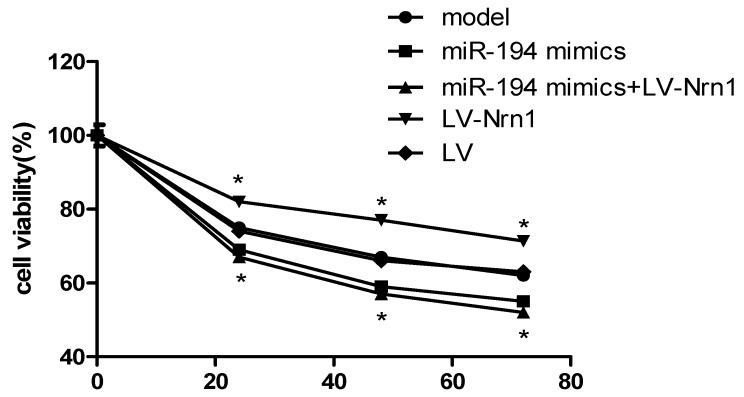
Nrn1 repaired the damage of miR-194 on Aβ_1–42-_transduced hippocampal neuron activity. * *p* < 0.05 when compared with the model group at the same time.

**Figure 7 genes-10-00313-f007:**
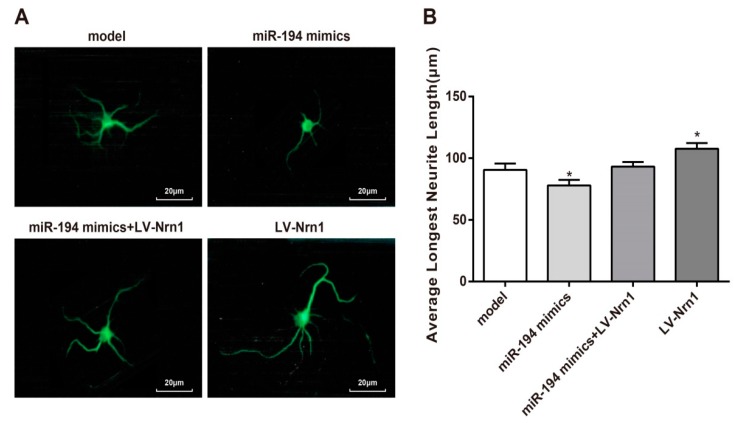
Nrn1 reversed the inhibitory effect of miR-194 on neurite growth of Aβ_1–42_-transduced hippocampal neurons. (**A**) Neurite of Aβ_1–42_-transduced hippocampal neurons were observed under a microscope after they were immunofluorescently stained. (**B**) Statistical data of the average longest neurite length. * *p* < 0.05 when compared with the model group at the same time.

**Figure 8 genes-10-00313-f008:**
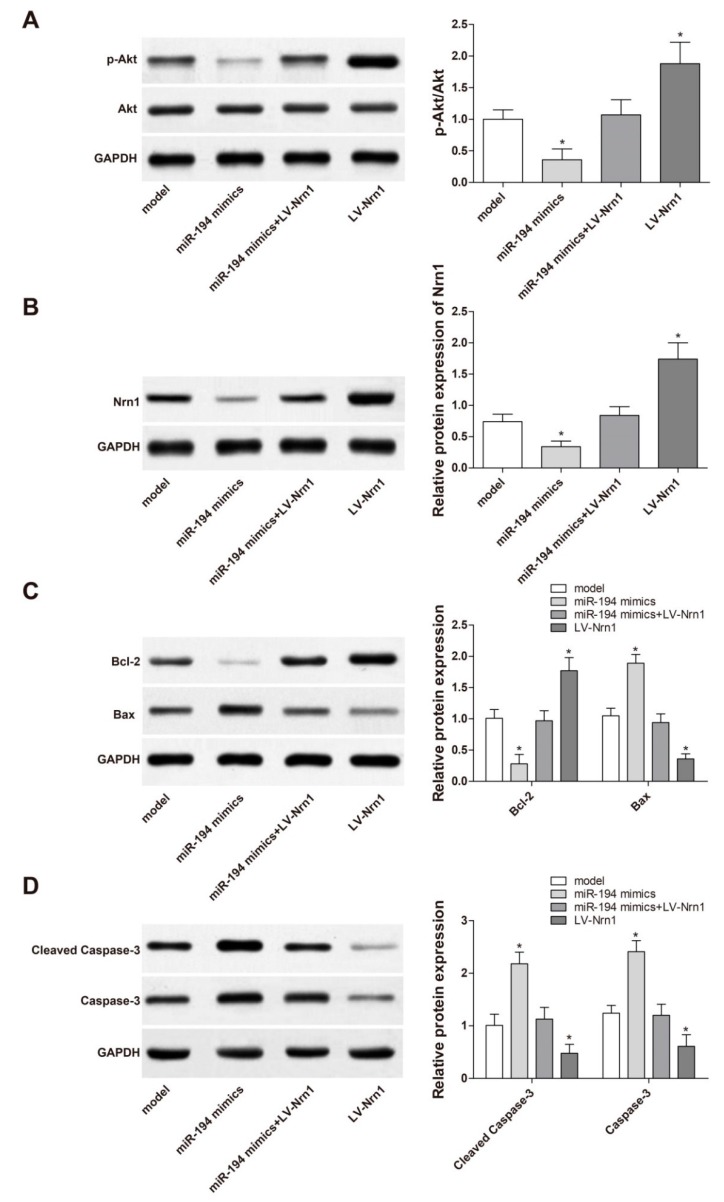
miR-194 regulated apoptosis-related protein expression in Aβ_1–42_-transduced hippocampal neurons by targeting inhibition of Nrn1 expression and decreasing PI3K/AkT signaling pathway activity. Western blot analysis of the expression of (**A**) AkT and p-AkT, (**B**) Nrn1, (**C**) Bcl-2 and Bax, and (**D**) Caspase-3 and Cleaved Caspase-3. * *p* < 0.05 when compared with the model group.

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
