# Peer review of "miR-194 Accelerates Apoptosis of Aβ1–42-Transduced Hippocampal Neurons by Inhibiting Nrn1 and Decreasing PI3K/Akt Signaling Pathway Activity"

_genes, 2019, doi:10.3390/genes10040313_

Round 1
Reviewer 1 Report
The manuscript, by T. Wang, et al., describes the effect of miR-194 on the proliferation and apoptosis after treatment of hippocampal neurons derived from rats and treated with Aβ1-42
Major points:
1. Abeta treatment of hippocampal neurons derived from probably wild type animals(rat genotype is not described) is not relevant experimental set up to study mechanisms the mechanisms of miR-194 on the proliferation and apoptosis of Alzheimer's disease (AD) hippocampal neurons, as it claimed in the manuscript.
2. The dose of Aβ1-42 was suboptimal, therefore effects may not be relevant to Alzheimer's disease. Appropriate controls (for example, treatment with scrambled Aβ1-42) was not used.
3. The study is not novel, the effect of Aβ1-42 on miR-194 has been previously described, a quick look in PubMed gives:
Lau et al., Alteration of the microRNA network during the progression of Alzheimer's disease. EMBO Mol Med. 2013 Oct; 5(10): 1613–1634.
Sørensen, et al., 2017 miRNA expression profiles in cerebrospinal fluid and blood of patients with Alzheimer’s disease and other types of dementia – an exploratory study.Transl Neurodegener. 2016; 5: 6.
Author Response
Dear Editors and Reviewers:
Thank you for your letter and for the reviewers’ comments concerning our manuscript entitled “miR-194 accelerates apoptosis of hippocampal neurons in Alzheimer's disease by targeting inhibition of Nrn1 and decreasing PI3K/Akt signaling pathway activity”. Revised portion are marked with traces in the paper.
Comments
Reviewer 1:
1. Abeta treatment of hippocampal neurons derived from probably wild type animals(rat genotype is not described) is not relevant experimental set up to study mechanisms the mechanisms of miR-194 on the proliferation and apoptosis of Alzheimer's disease (AD) hippocampal neurons, as it claimed in the manuscript.
Response: Thank you. Based on your professional comments, we have added the experiment as follows:
3.4 Up-regulated miR-194 inhibited hippocampal neurons viability and promoted its apoptosis
After 24-72 h of transfection, compared with model group, the viability of AD hippocampal neurons in miR-194 mimics group was significantly decreased (P < 0.05, Figure 2A).In addition, the apoptosis rates were all dramatically higher than those treated by miR-194 mimic transfection (P < 0.05, Figure 2B). These results indicated that the expression of miR-194 directly affected the proliferation and apoptosis of AD hippocampal neurons.
Figure 2 Up-regulated miR-194 decreased viability of AD hippocampal neurons and increased the apoptosis of AD hippocampal neurons. * P< 0.05 when compared with model group
2. The dose of Aβ1-42 was suboptimal, therefore effects may not be relevant to Alzheimer's disease. Appropriate controls (for example, treatment with scrambled Aβ1-42) was not used.
Response: Thank you for your professional comments. The dose we chosen was the Minimum effective dose after induced for 24h. 30μmol/L Aβ1-42 could inhibit hippocampal neurons viability and promote its apoptosis after induced for 24h. We have explained it in part 3.2. Based on your suggestion, we have added the scrambled Aβ1-42 and related experiments as follows:
Aβ1-42 peptide and its’ scrambled sequence (control) were synthesized by Shanghai Chu Peptide Biotechnology Co., Ltd. (Shanghai, China). Aβ1-42 peptide and its’ scrambled sequence (control) were synthesized by Shanghai Chu Peptide Biotechnology Co., Ltd. (Shanghai, China). 1 mg of Aβ1-42 monomer was dissolved in hexafluoroisopropanol to a concentration of 4.5 mg/ml, incubated for 60 min at room temperature, and hexafluoroisopropanol was evaporated under vacuum to add dimethyl sulfoxide to a concentration of 22 5 mg / ml, treated with ultrasonic bath for 10 min, added PBS containing 0.2% SDS, diluted to 2.5 mg / ml, incubated at 4 °C for 24 h, diluted with PBS to a final concentration, incubated at 4 °C for 2 weeks.
3. The study is not novel, the effect of Aβ1-42 on miR-194 has been previously described, a quick look in PubMed gives:
Lau et al., Alteration of the microRNA network during the progression of Alzheimer's disease. EMBO Mol Med. 2013 Oct; 5(10): 1613–1634.
Sørensen, et al., 2017 miRNA expression profiles in cerebrospinal fluid and blood of patients with Alzheimer’s disease and other types of dementia – an exploratory study.Transl Neurodegener. 2016; 5: 6.
Response: Thank you very much. In 2013, Pierre et al. [15] researched alteration of the microRNA network during the progression of Alzheimer’s disease, and found that miR-194 differently expressed in this disease. More importantly, miR-194-5p was confirmed to be significantly down-regulated in AD patients compared to controls [16]. However, specific mechanisms of miR-194 such as related targets and pathways of AD has have not been studied. In this study, we aimed to research these specific mechanisms. We have added these contents in the introduction.
Hope the revised manuscript could be satisfied to you. If there are any other questions, please contact us with no hesitate.
Best wishes.
Xiaocui Liu

Reviewer 2 Report
In this study the authors exposed rodent hippocampal neurons developed using standard in vitro culture techniques to Abeta (1-42) peptide and found a decrease in Neuritin-1 mRNA and increase in miR-194 microRNA expression. The then used vectors that "mimic" miR-194 and/or increase expression of Neuritin-1 (Nrn-1) mRNA and showed inverse relationships (ie, miR-194 decreased expression of Nrn-1 and vice-versa). They examined various proteins associated with cell growth (Akt), cell death (caspase-3, bcl-2), or fluorescent markers for mitochondrial membrane potential (Rhodamine123) and found that miR-194 reduced Akt expression, increased cell death markers, reduced mitochondrial membrane potential. All of these were reversed by increased expression of Nrn-1.
The findings wrt miR-194 and Nrn-1 are interesting and potentially useful to advancing the field. However, the paper is replete with incomplete descriptions of Methods and poorly written wrt English spelling and style. These deficits will need addressing before publication.
I have the following concerns:
There are multiple spelling errors and more importantly, deficits in English style. In some cases, the paper is very difficult to read and/or interpret. A native English speaker should revise the paper.
There are hundreds of papers using rodent hippocampal neurons in culture and dozens of papers showing the Abeta peptides are neurotoxic in the uM concentration range. Thus, these findings are far from novel and can be described. The Figures showing these findings can be presented as Supplemental data to document the cell model used.
No details are provided as to how the Abeta(1-42) peptide was prepared. This particular Abeta analogue is known to self-aggregate, and it is important that the authors not only describe how this peptide was prepared, but document its aggregation state. I doubt that any of the Abeta (1-42) was monomeric, but rather was oligomeric or extensively aggregated.
Figure 1 can be in Supplemental data. The bottom right panel of Figure 1 needs to have the green fluorescence reduced and/or blue nuclear fluorescence increased. The "overlap" between the two is impossible to discern.
Likewise, Figure 2 can be in Supplemental data, as there is nothing new here. Again, knowledge of the preparation of Abeta (1-42) is essential.
Figure 3. What is the duration of incubation with Abeta (1-42) peptide before the neurons were harvested and assayed with qPCR? What is "blank" (reverse peptide? scrambled peptide? buffer?)
Figure 4. No criticisms. This is important data for the thrust of the paper. The authors might indicate why they looked for miR-194 among the thousands of microRNAs. Did they perform a miRNA screen?
Figure 5. What is "LV-Nrn1"? Although they purchased the expression vectors, they should provide details of what they are. I'm assuming they were based on a lentiviral backbone, as they were transfecting neurons. In addition, Fig.5 shows that LV-Nrn-1 is toxic to their neurons. Thus, it is very important that the authors describe their expression vector. Did they carry out any empty vector controls? If not, why? Does LV-Nrn-1 increase expression of Nrn-1 (Neuritin-1)? If so, this needs to be demonstrated (can be in Supplemental data)
Figure 6. Neuritin-1 is known to increase neurite length, so Figure 6 is superfluous unless the authors demonstrate that their expression vector increases Neuritin-1 protein. Again, details of transfection are critical.
Figure 7. The authors used rhodamine123 as a semi-quantitative marker of mitochondrial membrane potential. Rh123 is known to exhibit self-quenching, which is evidently the property that the authors depended on to generate Fig. 7. I suggest they do one of two things: either include a depolarization control (ie FCCP) or use a different probe, like TMRE or JC-1.
Figure 8. This is an important figure to their argument. Transfection details need to be provided.
In summary, this is a potentially contributory paper to AD research. The findings wrt miR-194 and Nrn-1 are novel and worth reporting. The "model" they used has extensive historical precedent but may not be relevant to sporadic AD. The authors at least need to mention this alternative view in their Discussion.
Author Response
Dear Editors and Reviewers:
Thank you for your letter and for the reviewers’ comments concerning our manuscript entitled “miR-194 accelerates apoptosis of hippocampal neurons in Alzheimer's disease by targeting inhibition of Nrn1 and decreasing PI3K/Akt signaling pathway activity”. Revised portion are marked with traces in the paper.
Comments
Reviewer 2:
1. There are multiple spelling errors and more importantly, deficits in English style. In some cases, the paper is very difficult to read and/or interpret. A native English speaker should revise the paper.
Response: We were sorry for the poor level of our English. We have revised the manuscript with the help of a native speaker. Hope the revised manuscript could be satisfied to you.
2. There are hundreds of papers using rodent hippocampal neurons in culture and dozens of papers showing the Abeta peptides are neurotoxic in the uM concentration range. Thus, these findings are far from novel and can be described. The Figures showing these findings can be presented as Supplemental data to document the cell model used.
Response:Thank you for your professional comments. As you say, the neurotoxicity of Abeta peptides was not research focus in this study. We have presented those results as Supplemental data.
3. No details are provided as to how the Abeta(1-42) peptide was prepared. This particular Abeta analogue is known to self-aggregate, and it is important that the authors not only describe how this peptide was prepared, but document its aggregation state. I doubt that any of the Abeta (1-42) was monomeric, but rather was oligomeric or extensively aggregated.
Response: Thank you for your comment. We have revised the method as follows: Aβ1-42 peptide and its’ scrambled sequence (control) were synthesized by Shanghai Chu Peptide Biotechnology Co., Ltd. (Shanghai, China). 1 mg of Aβ1-42 monomer was dissolved in hexafluoroisopropanol to a concentration of 4.5 mg/ml, incubated for 60 min at room temperature, and hexafluoroisopropanol was evaporated under vacuum to add dimethyl sulfoxide to a concentration of 22 5 mg / ml, treated with ultrasonic bath for 10 min, added PBS containing 0.2% SDS, diluted to 2.5 mg / ml, incubated at 4 °C for 24 h, diluted with PBS to a final concentration, incubated at 4 °C for 2 weeks.
4. Supplemental Figure 1 can be in Supplemental data. The bottom right panel of Supplemental Figure 1 needs to have the green fluorescence reduced and/or blue nuclear fluorescence increased. The "overlap" between the two is impossible to discern.
Response: Thank you. We have revised Figure 1 as supplemental Figure 1, and overlap was also revised.
5. Likewise, Figure 2 can be in Supplemental data, as there is nothing new here. Again, knowledge of the preparation of Abeta (1-42) is essential.
Response: Thank you very much. Knowledge of the preparation of Abeta (1-42) has been added, and figure 2 was as supplemental data.
6. Figure 3. What is the duration of incubation with Abeta (1-42) peptide before the neurons were harvested and assayed with qPCR? What is "blank" (reverse peptide? scrambled peptide? buffer?)
Response: Thank you very much. After induced by 30 μmol/L Aβ1-42 for 48h, miR-194 relative levels in AD hippocampal neurons of Aβ1-42 group was much higher than that of Blank group (P < 0.05). The scrambled peptide was used for the black group. In contrast, Nrn1 mRNA was markedly increased in AD hippocampal neurons of Aβ1-42 group when compared with that of Blank group. We have added above contents in manuscript.
7. Figure 4. No criticisms. This is important data for the thrust of the paper. The authors might indicate why they looked for miR-194 among the thousands of microRNAs. Did they perform a miRNA screen?
Response: Thank you. miR-194 was also reported to be involved in the development and progression of many diseases, such as tumors and inflammatory reactions. In 2013, Pierre et al. [15] researched alteration of the microRNA network during the progression of Alzheimer’s disease, and found that miR-194 differently expressed in this disease. More importantly, miR-194-5p was confirmed to be significantly down-regulated in AD patients compared to controls [16]. However, specific mechanisms of miR-194 such as related targets and pathwaysthe effect of miR-194 on of AD has have not been studied. We have added these contents in introduction.
8. Figure 5. What is "LV-Nrn1"? Although they purchased the expression vectors, they should provide details of what they are. I'm assuming they were based on a lentiviral backbone, as they were transfecting neurons. In addition, Fig.5 shows that LV-Nrn-1 is toxic to their neurons. Thus, it is very important that the authors describe their expression vector. Did they carry out any empty vector controls? If not, why? Does LV-Nrn-1 increase expression of Nrn-1 (Neuritin-1)? If so, this needs to be demonstrated (can be in Supplemental data)
Response: Thank you. Lentiviral vectors were purchased from Tronolab, and the packaging cell 293T cell line was purchased from the Shanghai Institute of Cellular Sciences, Chinese Academy of Sciences. We have added it in part 2.7. The empty vector controls was also added (as following figure). In the Pre-test, LV-Nrn-1 increased expression of Nrn-1.
9. Figure 6. Neuritin-1 is known to increase neurite length, so Figure 6 is superfluous unless the authors demonstrate that their expression vector increases Neuritin-1 protein. Again, details of transfection are critical.
Response: Thank you. In this study, we aimed to research whether miR-194 mimic could decrease neurite length, and Neuritin-1 could restore the affection of miR-194 mimic. Thereby, we hope to keep this Figure. Details of transfection were added as follows:
Briefly, hippocampal neurons cells were plated in 6-well plates at a density of 5x105/well and allowed to adhere for 24 h. 5 µl Lipofectamine 2000 with miR-194 mimics and/or Nrn1 overexpression vectors were used per well.
10. Figure 7. The authors used rhodamine123 as a semi-quantitative marker of mitochondrial membrane potential. Rh123 is known to exhibit self-quenching, which is evidently the property that the authors depended on to generate Fig. 7. I suggest they do one of two things: either include a depolarization control (ie FCCP) or use a different probe, like TMRE or JC-1.
Response: Thank you for your professional comments. As you say, Rh123 is known to exhibit self-quenching. Including a depolarization control (ie FCCP) or use a different probe, like TMRE or JC-1, is better. However, the experiments can’t be processed based on the experiment condition. In order to make up this deficiency, flow cytometry for Mitochondrial membrane potential was added as follows:
1 ml of the prepared single cell suspension was taken. These cells were washed with cold PBS, and incubated with 100 ul of PBS. Rhodamine 123 dye was added to a final concentration of 10 ug / ml. At 37°C, the light was equilibrated for 30 minutes, the cells were washed with PBS, and the mitochondrial membrane potential was measured by flow cytometry. The results were expressed as mean fluorescence intensity.
11. Figure 8. This is an important figure to their argument. Transfection details need to be provided.
Response: Thank you. The transfection detail was added as follows:
AD hippocampal neurons induced by 30 μmol/L of Aβ1-42 were collected and transfected by miR-194 mimics (miR-194 mimics group) or Nrn1 overexpression vectors (LV-Nrn1 group). In addition, they were also co-transfected with both miR-194 mimics and Nrn1 overexpression vectors (lentiviral backbone), and were served as miR-194 mimics+ LV-Nrn1 group. Lentiviral vectors were purchased from Tronolab, and the packaging cell 293T cell line was purchased from the Shanghai Institute of Cellular Sciences, Chinese Academy of Sciences. Furthermore, AD hippocampal neurons only induced by 30 μmol/L of Aβ1-42 were set as model group. miR-194 mimics and Nrn1 overexpression vectors were purchased from Guangzhou Ruibo Biotechnology Co., Ltd., China. Transfection was performed in strict accordance with the instructions of Lipofectamine® 2000. Briefly, hippocampal neurons cells were plated in 6-well plates at a density of 5x105/well and allowed to adhere for 24 h. 5 µl Lipofectamine 2000 with miR-194 mimics and/or Nrn1 overexpression vectors were used per well. After transfection, all AD hippocampal neurons were cultured in the 37°C, 5% CO2 incubator for 0-72 h. Cell viability was also detected using MTT assay as above.
12. In summary, this is a potentially contributory paper to AD research. The findings wrt miR-194 and Nrn-1 are novel and worth reporting. The "model" they used has extensive historical precedent but may not be relevant to sporadic AD. The authors at least need to mention this alternative view in their Discussion.
Response: Thank you. Based on your suggestion, we have mentioned it in limitations.
Hope the revised manuscript could be satisfied to you. If there are any other questions, please contact us with no hesitate.
Best wishes.
Xiaocui Liu

Reviewer 3 Report
Wang et al. studied the effect of microRNA miR-194 on cultured rat hippocampal neurons challenged with AB1-42, as a model of AD. The authors identified Nrn1 as a target gene, and compared multiple phenotypes after genetic perturbation. However, a few minor problems must be addressed before publication:
In Rhodamine 123 results (3.7), the authors only showed one representative image for each experimental group. Quantification and statistical analysis should be performed.
The procedure of computational miR-194 target scan (leading to results in 3.4 and Fig. 4A) should be described in the methods section.
The statistical section (2.10) should state whether t tests were one-tailed or two-tailed.
P-values in the results section should be exact values instead of just "P < 0.05". However, this should depend on journal policy.
The subheading of 3.4 should be "a target gene" instead of "the target gene", since the authors cannot rule out other miR-194 target genes.
The words "repair the damage" in 3.5 are not accurate, because the authors did not cause damage with miR-194 first and then use Nrn1 to "repair" it.
Many typos (or grammar and formatting mistakes) must be fixed throughout the manuscript. Some severe ones include: in introduction "discovered to be normally expressed" should be "abnormally"; in 2.3 "1x104/well" should have "4" as superscript; in 2.4 "LgG" should be "IgG".
Author Response
Dear Editors and Reviewers:
Thank you for your letter and for the reviewers’ comments concerning our manuscript entitled “miR-194 accelerates apoptosis of hippocampal neurons in Alzheimer's disease by targeting inhibition of Nrn1 and decreasing PI3K/Akt signaling pathway activity”. Revised portion are marked with traces in the paper.
Comments
Reviewer 3:
1. In Rhodamine 123 results (3.7), the authors only showed one representative image for each experimental group. Quantification and statistical analysis should be performed.
Response: Thank you for your professional comments. We have added the quantification and statistical analysis as follows:
1 ml of the prepared single cell suspension was taken. These cells were washed with cold PBS, and incubated with 100 ul of PBS. Rhodamine 123 dye was added to a final concentration of 10 ug / ml. At 37°C, the light was equilibrated for 30 minutes, the cells were washed with PBS, and the mitochondrial membrane potential was measured by flow cytometry. The results were expressed as mean fluorescence intensity.
Rhodamine 123 was a fluorescent dye that could pass through cell membranes to selectively stain live cell mitochondria, which was widely used as a fluorescent probe for detecting mitochondrial membrane potential. As shown in Figure 6, the green fluorescence intensity of miR-194 mimics group was significantly stronger than that of model group, while LV-Nrn1 group had weaker fluorescence intensity than that of model group. It could also be noticed that, miR-194 mimics + LV-Nrn1 group had similar green fluorescence intensity to model group, illustrating that Nrn1 inhibited the decrease of mitochondrial transmembrane potential in AD hippocampal neurons induced by miR-194. Based on the result of mean fluorescence intensity, the similar result was also obtained.
2. The procedure of computational miR-194 target scan (leading to results in 3.4 and Fig. 4A) should be described in the methods section.
Response: Target Scan was an online prediction software to predict the binding site of miRNAs and genes. Based on the database, miR-194 was direct binding to the 3'-UTR region of Nrn1 according to the speculation of Target Scan (Figure 4A). We have added the content.
3. The statistical section (2.10) should state whether t tests were one-tailed or two-tailed.
Response: Thank you. Two-tailed t tests was used in this study.
4. P-values in the results section should be exact values instead of just "P < 0.05". However, this should depend on journal policy.
Response: Thank you. In journal policy, P-values should be shown as P < 0.05.
5. The subheading of 3.4 should be "a target gene" instead of "the target gene", since the authors cannot rule out other miR-194 target genes.
Response: We were sorry for our mistakes. We have revised it to "a target gene"
6. The words "repair the damage" in 3.5 are not accurate, because the authors did not cause damage with miR-194 first and then use Nrn1 to "repair" it.
Response: We were sorry for our mistakes. We have revised it to “resume”
7. Many typos (or grammar and formatting mistakes) must be fixed throughout the manuscript. Some severe ones include: in introduction "discovered to be normally expressed" should be "abnormally"; in 2.3 "1x104/well" should have "4" as superscript; in 2.4 "LgG" should be "IgG".
Response: Thank you. We were sorry for our mistakes. We have checked the manuscript and corrected the errors.
Hope the revised manuscript could be satisfied to you. If there are any other questions, please contact us with no hesitate.
Best wishes.
Xiaocui Liu

Round 2
Reviewer 1 Report
Dear Authors,
thank you for your efforts, however, my main concerns are the same, that the experimental setup and lack of proper controls do not support the conclusion drawn.
The authors mentioned: "Aβ1-42 was commonly used to establish AD cell models [18,19]"
First, references 18 and 19 do not support that the treatment with Abeta is used to establish AD cell models.
ref 18: SH-SY5Y cells overexpressing human β-amyloid precursor protein (APP) and Aβ1-42-transgenic C. elegans GMC101.
ref 19, injection of Ab42 to the brain was used to model AD.
Second, I do not agree with the statement that "Aβ1-42 at a density of 30 μmol/L could induce stable AD hippocampal neurons model". "stable" means that the acquired properties could be propagated through several cell generations, which is not possible to achieve by treatment with Abets.
Authors may read the paper:
Laura Rodríguez-Pascau et al., 2012 "Generation of a Human Neuronal Stable Cell Model for Niemann-Pick C Disease by RNA Interference"
Treatment cells with Aβ1-42 was described in ref. [9, 10, 14].
2. Appropriate controls (for example, treatment with scrambled Aβ1-42) was not used.
Authors added scrambled Aβ1-42 in Material and Methods however, they do not provide data showing that scrambled Aβ1-42 was used in the experiments.
Author Response
Dear Editors and Reviewers:
Thank you for your letter and for the reviewers’ comments concerning our manuscript entitled “miR-194 accelerates apoptosis of hippocampal neurons in Alzheimer's disease by inhibiting Nrn1 and decreasing PI3K/Akt signaling pathway activity”. Revised portion are marked with traces in the paper.
Comments
Reviewer 1:
1. thank you for your efforts, however, my main concerns are the same, that the experimental setup and lack of proper controls do not support the conclusion drawn.
The authors mentioned: "Aβ1-42 was commonly used to establish AD cell models [18,19]"
First, references 18 and 19 do not support that the treatment with Abeta is used to establish AD cell models.
ref 18: SH-SY5Y cells overexpressing human β-amyloid precursor protein (APP) and Aβ1-42-transgenic C. elegans GMC101.
ref 19, injection of Ab42 to the brain was used to model AD.
Response: Thank you for your comment. We apologize for the errors we have made. We have revised and marked with traces in the manuscript as following: “It has been reported that Aβ1-42 is used to establish AD cell modles [17, 19]”. In addition, reference 19 was revised as follows:
[17] Geng D., Kang L., Su Y., et al., Protective effects of EphB2 on Abeta1-42 oligomer-induced neurotoxicity and synaptic NMDA receptor signaling in hippocampal neurons. Neurochem Int 2013, 63, 283-90, 10.1016/j.neuint.2013.06.016.
[19] Zhang M., Liu Y., Liu M., et al., UHPLC-QTOF/MS-based metabolomics investigation for the protective mechanism of Danshen in Alzheimer's disease cell model induced by Abeta1-42. Metabolomics 2019, 15, 13, 10.1007/s11306-019-1473-x.
2. Second, I do not agree with the statement that "Aβ1-42 at a density of 30 μmol/L could induce stable AD hippocampal neurons model". "stable" means that the acquired properties could be propagated through several cell generations, which is not possible to achieve by treatment with Abets.
Authors may read the paper:
Laura Rodríguez-Pascau et al., 2012 "Generation of a Human Neuronal Stable Cell Model for Niemann-Pick C Disease by RNA Interference"
Treatment cells with Aβ1-42 was described in ref. [9, 10, 14].
Response: Thank you for your professional comments. In our article, we would like to say that Aβ1-42 at a density of 30 μmol/L could successfully induce AD cell model. We have revised and marked with traces in the manuscript as following: “Aβ1-42 at a density of 30 μmol/L could successfully induce AD cell model.”
3. Appropriate controls (for example, treatment with scrambled Aβ1-42) was not used.
Authors added scrambled Aβ1-42 in Material and Methods however, they do not provide data showing that scrambled Aβ1-42 was used in the experiments.
Response: Thank you for your comment. As described in part 2.3 of material and methods, “Aβ1-42 peptide and its’ scrambled sequence (control) synthesized by Shanghai Chu Peptide Biotechnology Co., Ltd., were respectively as follows: Aβ1-42, DAEFRHDSGYEVHHQKLVFFAEDVGSNKGAIIGLMVGGVVIA; scrambled Aβ1-42, AIAEGDSHVLKEGAYMEIFDVQGHVFGGKIFRVVDLGSHNVA.” In results part, we just used control instead of scrambled Aβ1-42 peptide sequence to illustrate our results in Figure 2. Thank you again for your comments.
Hope the revised manuscript could be satisfied to you. If there are any other questions, please contact us with no hesitate.
Best wishes.
Xiaocui Liu

Reviewer 2 Report
The authors have addressed most of my scientific concerns. The presentation of mitochondrial membrane potential assay remains inadequate and can be removed. There remain multiple English language editing deficits that need addressing. I still cannot find where the authors discuss how their rodent hippocampal neuron exposed to Abeta peptides has an extensive history in the literature and questionably relates to sporadic AD. Also, the authors have improved the description of how they prepared Abeta peptides but do not provide any evidence of the state of aggregation. This point is very important, as oligomeric (not extensively aggregated) Abeta is thought to be the most toxic species.
Author Response
Dear Editors and Reviewers:
Thank you for your letter and for the reviewers’ comments concerning our manuscript entitled “miR-194 accelerates apoptosis of hippocampal neurons in Alzheimer's disease by inhibiting Nrn1 and decreasing PI3K/Akt signaling pathway activity”. Revised portion are marked with traces in the paper.
Comments
Reviewer 2:
1. The authors have addressed most of my scientific concerns. The presentation of mitochondrial membrane potential assay remains inadequate and can be removed.
Response: Thank you. Based on your professional comments, we have removed the mitochondrial membrane potential assay.
2. There remain multiple English language editing deficits that need addressing.
Response: We were sorry for the editing errors in the manuscript. We have revised and checked the manuscript. Hope the revised manuscript could be satisfied to you.
3. I still cannot find where the authors discuss how their rodent hippocampal neuron exposed to Abeta peptides has an extensive history in the literature and questionably relates to sporadic AD.
Response: Thank you for your comment. Based on your professional comments, we have revised and marked with traces in the discussion section of our manuscript as following: “The AD cell model used in this study has extensive historical precedent. Previous studies have reported that control of ERK signaling might be useful in preventing Aβ1-42_oligomer-induced neurotoxicity in the hippocampus [34]. Researches also have found that EphB2 overexpression protects hippocampal neurons against Aβ1-42 oligomer-induced neurotoxicity in the cellular model of AD [17].”
4. Also, the authors have improved the description of how they prepared Abeta peptides but do not provide any evidence of the state of aggregation. This point is very important, as oligomeric (not extensively aggregated) Abeta is thought to be the most toxic species.
Response: Thank you for your comment. Based on your professional comments, we have revised and marked with traces in the part 2.3 of our manuscript as following: “Aβ1-42 oligomers were prepared as previously described [17]”. The following literatures are available for reference:
Zhang M., Liu Y., Liu M., et al., UHPLC-QTOF/MS-based metabolomics investigation for the protective mechanism of Danshen in Alzheimer's disease cell model induced by Abeta1-42. Metabolomics 2019, 15, 13, 10.1007/s11306-019-1473-x.
Geng D., Kang L., Su Y., et al., Protective effects of EphB2 on Abeta1-42 oligomer-induced neurotoxicity and synaptic NMDA receptor signaling in hippocampal neurons. Neurochem Int 2013, 63, 283-90, 10.1016/j.neuint.2013.06.016.
Chong Y.H., Shin Y.J., Lee E.O., et al., ERK1/2 activation mediates Abeta oligomer-induced neurotoxicity via caspase-3 activation and tau cleavage in rat organotypic hippocampal slice cultures. J Biol Chem 2006, 281, 20315-25, 10.1074/jbc.M601016200.
Hope the revised manuscript could be satisfied to you. If there are any other questions, please contact us with no hesitate.
Best wishes.
Xiaocui Liu

Round 3
Reviewer 1 Report
Dear Authors,
Except for the major comment that experimental setup is not relevant to AD, since the concentrations of Aβ used for this study are much above these which could be relevant to AD (sub-micromolar range), I have only a few minor issues:
I would recommend to use labeling instead of staining ("Staining" is traditionally referred to the use of dyes, "labeling" is used for antibodies).
line 220 Figure 1. Immunofluorescence labeling, please indicate antibodies used.
“treated by 0 μmol/L Aβ" seems to be unusual in this context.
Figure 2.”1-42" should be in subscript
Author Response
Dear Editors and Reviewers:
Thank you for your letter and for the reviewers’ comments concerning our manuscript entitled “miR-194 accelerates apoptosis of Aβ1-42 transduced hippocampal neurons by inhibiting Nrn1 and decreasing PI3K/Akt signaling pathway activity”. Revised portion are marked with traces in the paper.
1. Except for the major comment that experimental setup is not relevant to AD, since the concentrations of Aβ used for this study are much above these which could be relevant to AD (sub-micromolar range), I have only a few minor issues:
Response: Thank you for your comment. The high concentrations of Aβ1-42 used for our study may be due to the purity of purchased Aβ1-42. Thanks for your comment again.
2. I would recommend to use labeling instead of staining ("Staining" is traditionally referred to the use of dyes, "labeling" is used for antibodies).
Response: Thank you for your comment. We have revised “staining” to “labeling”. Thanks for your valuable comment again.
3. line 220 Figure 1. Immunofluorescence labeling, please indicate antibodies used.
Response: Thank you for your comment. We have revised and marked with traces in the manuscript as following: “(E and F) Immunofluorescence labeling of hippocampal neurons treated by mouse anti-rat β-III tubulin monoclonal antibody and FITC-labeled goat anti-mouse IgG secondary antibody.”. Thanks for your valuable comment again.
4. “treated by 0 μmol/L Aβ" seems to be unusual in this context.
Response: Thank you for your comment. We set 0 μmol/L Aβ1-42 group to confirmed that the solution in which Aβ1-42 was dissolved did not affect the experimental results and 0 μmol/L Aβ1-42 group was described as negative controls in the revised manuscript.
5. Figure 2.”1-42" should be in subscript
Response: Thank you for your comment. We have revised and checked the manuscript. Hope the revised manuscript could be satisfied to you.
Hope the revised manuscript could be satisfied to you. If there are any other questions, please contact us with no hesitate.
Best wishes.
Xiaocui Liu